

# Magpie Monitor

## System raportowania zdarzeń systemowych w języku naturalnym z wykorzystaniem dużych modeli językowych

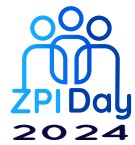

**Autors**: Marcel Dybek ⊙ · Adam Lamers ⊙ · Wojciech Suszko ⊙ · Nikodem Świerkowski ⊙

**Supervisor:** Marcin Pietranik

### Abstract

Celem projektu jest zaprojektowanie i wdrożenie zaawansowanego systemu monitorowania oraz analizy logów generowanych przez klaster komputerowy, mającego na celu poprawę efektywności zarządzania infrastrukturą IT oraz minimalizację ryzyka przestojów aplikacji działających w klastrze.

Tradycyjne podejście do monitorowania aplikacji, oparte na ręcznej analizie logów jest czasochłonne i podatne na błędy. Opracowywany system automatyzuje ten proces, znacząco redukując ilość danych wymagających przeglądu przez administratorów oraz dostarczając inteligentne rekomendacje dotyczące dalszych działań. Dzięki temu administratorzy mogą skupić się na analizie kluczowych anomalii wykrywanych przez system, zamiast poświęcać czas na manualne przetwarzanie dużych zbiorów logów.

Kluczowym elementem projektu jest zastosowanie dużego modelu językowego, który generuje raporty w języku naturalnym. Raporty te podsumowują najważniejsze zdarzenia systemowe w określonym przedziale czasowym, dostosowanym do potrzeb administratorów. Rozwiązanie to nie tylko zwiększa efektywność monitorowania systemów, ale również ułatwia identyfikację i zrozumienie zarówno problemów krytycznych, jak i niekrytycznych, wspierając szybkie oraz skuteczne podejmowanie decyzji.

## 1    WPROWADZENIE

W dobie rozbudowanych aplikacji webowych, które muszą działać nieprzerwanie przez całą dobę, monitorowanie aktualnego stanu systemu jest kluczowe dla zapewnienia ciągłości działania oraz wysokiej jakości doświadczeń użytkowników końcowych.

Obecnie stosowane rozwiązania monitorujące logi opierają się głównie na wyświetlaniu oraz zliczaniu, co umożliwia administratorom podsumowanie stanu monitorowanej aplikacji. Często wykorzystywana jest również analiza słów kluczowych lub wzorców, które pomagają zidentyfikować najbardziej znaczące zapisy dotyczące działania systemu. Takie podejście jest czasochłonne i działa już po wystąpieniu awarii. Nie wyróżnia logów mogących sygnalizować przyszłe, potencjalne problemy, skupiając się głównie na wykrywaniu już istniejących awarii.

Celem opracowanego systemu jest projekt i implementacja rozwiązania, które będzie nieustannie zbierać dane generowane przez monitorowany system, analizować je w zadanym przedziale czasu i generować raporty dotyczące kluczowych incydentów. System ten pozwoli na identyfikację istotnych, lecz często pomijanych informacji, dzięki czemu administratorzy będą mogli regularnie monitorować stan systemu nawet w przypadku ogromnej ilości generowanych logów. W rezultacie, takie podejście powinno zminimalizować liczbę awarii, skrócić czas potrzebny na naprawę problemów oraz obniżyć koszty związane z administrowaniem infrastrukturą IT, za sprawą mniejszej inwestycji czasowej administratorów.

Osiągnięcie tego, wymaga dostarczenia klientowi agenta, który zainstalowany na wybranym klastrze komputerowym, będzie zbierać generowane logi oraz wysyłać je do analizy. W konsekwencji system dostarczany jest w postaci SaaS (oprogramowanie jako usługa), wymagając od klienta jedynie zainstalowania agenta zbierającego logi z jego klastra komputerowego.

Kluczowe jest także zapewnienie użytkownikom intuicyjnego interfejsu oraz możliwości powiadamiania administratorów o zdarzeniach systemowych za pomocą zewnętrznych kanałów komunikacyjnych.

## 2 POWIĄZANE PRACE

### 2.1 ScienceLogic

ScienceLogic [3] to zaawansowane narzędzie monitorujące, przeznaczone do kompleksowego zarządzania infrastrukturą IT i aplikacjami. Wykorzystuje automatyzację operacji IT (AIOps), analizując w czasie rzeczywistym zależności między komponentami systemu, co umożliwia szybkie identyfikowanie problemów i ich przyczyn. ScienceLogic pozwala na integrację z wieloma platformami oraz wsparcie dla różnorodnych środowisk, takich jak chmura, serwery lokalne czy kontenery. W porównaniu do Magpie Monitor, wykorzystuje model sztucznej inteligencji już po wystąpieniu błędu, aby jak najszybciej znaleźć jego przyczynę, zamiast skupiać się na próbie znalezienia błędu nim wywoła on awarię. ScienceLogic posiada szerszy zakres funkcji monitorujących, jednak brakuje mu dedykowanego generowania raportów w języku naturalnym oraz rekomendacji działań opartych na zaawansowanych modelach językowych, co sprawia, że Magpie Monitor może być bardziej przystępny dla użytkowników, którzy potrzebują prostszego systemu do analizy logów i raportowania.

### 2.2 Elasticsearch

Elasticsearch [2] to wszechstronne narzędzie do przeszukiwania i analizy danych w czasie rzeczywistym, powszechnie wykorzystywane jako podstawa systemów logowania i monitorowania. Jego zdolność do szybkiego indeksowania oraz przeszukiwania dużych zbiorów danych sprawia, że jest ceniony w środowiskach o dużym natężeniu informacji. W połączeniu z narzędziami takimi jak Kibana, Elasticsearch umożliwia wizualizację trendów i incydentów, co znacząco wspiera proces analizy systemów IT. Jednym z jego zaawansowanych rozszerzeń jest wtyczka opracowana przez Elastic Observability Labs, wykorzystująca mechanizm Watcher. Umożliwia ona analizę logów aplikacji i reagowanie na zdarzenia, w tym integrację z modelem GPT-4 od OpenAI, co dodatkowo zwiększa możliwości interpretacji logów i automatyzacji reakcji. W porównaniu do Magpie Monitor, Elasticsearch oferuje większą elastyczność w tworzeniu dostosowanych do potrzeb rozwiązań monitorujących. Mechanizm Watcher pozwala na definiowanie zaawansowanych reguł, które nie tylko wysyłają powiadomienia, ale także uruchamiają skrypty automatyzujące reakcję na zdarzenia. Jednak konfiguracja i pełne wykorzystanie jego funkcjonalności wymaga wiedzy technicznej oraz czasochłonnej personalizacji. Magpie Monitor natomiast wyróżnia się gotowym do użycia podejściem, skupionym na intuicyjnej obsłudze i generowaniu zautomatyzowanych, ustrukturyzowanych raportów w języku naturalnym. To sprawia, że jest bardziej przyjazny dla użytkowników, którzy potrzebują szybkiego dostępu do kluczowych informacji bez konieczności zaawansowanej konfiguracji. W efekcie, choć Elasticsearch z Watcherem jest bardziej elastyczny i potężny w środowiskach wymagających zaawansowanej analityki, Magpie Monitor oferuje większą prostotę i skuteczność w codziennym wykrywaniu logami i incydentami.

### 2.3 Log Analyzer

Log Analyzer [1] to proste, ale skuteczne narzędzie do analizy logów, zaprojektowane głównie z myślą o identyfikacji błędów oraz monitorowaniu wydajności systemu. Wykorzystuje model Chat GPT 4.0-mini z dodatkowym kontekstem, co pozwala na lepsze dopasowanie analizy do specyfiki zdarzeń systemowych w porównaniu do standardowego modelu. System cechuje się łatwością konfiguracji i użytkowania, co czyni go atrakcyjnym rozwiązaniem dla mniej złożonych środowisk. Jego możliwości są jednak ograniczone do podstawowej analizy i agregacji logów, co sprawia, że nie spełnia wymagań bardziej zaawansowanych operacji monitorujących. W porównaniu do Magpie Monitor, Log Analyzer wypada mniej wszechstronnie. Nie oferuje zaawansowanej automatyzacji ani możliwości generowania szczegółowych raportów w języku naturalnym, które podsumowują kluczowe incydenty i rekomendują działania. Magpie Monitor przewyższa to narzędzie również pod względem architektury – zastosowanie mikroserwisów pozwala na wysoką skalowalność i efektywność w zarządzaniu dużymi wolumenami danych z różnych źródeł. Dzięki temu Magpie Monitor jest bardziej kompleksowym i elastycznym rozwiązaniem, dostosowanym do potrzeb nowoczesnych środowisk IT.

### 2.4 Porównanie

W porównaniu do Magpie Monitor, Log Analyzer wypada mniej wszechstronnie. Nie oferuje zaawansowanej automatyzacji ani możliwości generowania szczegółowych raportów w języku naturalnym, które podsumowują kluczowe incydenty i rekomendują działania. Magpie Monitor przewyższa to narzędzie również pod względem architektury – zastosowanie mikroserwisów pozwala na wysoką skalowalność i

efektywność w zarządzaniu dużymi wolumenami danych z różnych źródeł. Dzięki temu Magpie Monitor jest bardziej kompleksowym i elastycznym rozwiązaniem, dostosowanym do potrzeb nowoczesnych środowisk IT.

# 3 UŻYTE TECHNOLOGIE

## 3.1 Kubernetes

Najbardziej dojrzały i powszechnie stosowany orkiestrator rozproszonych systemów opartych na kontenerach, szeroko wykorzystywany w zastosowaniach komercyjnych.

## 3.2 Docker

Najpopularniejsze narzędzie i ekosystem do budowania oraz uruchamiania kontenerów aplikacji.

## 3.3 Golang

Język programowania umożliwiający tworzenie szybkich, odpornych na wycieki pamięci i wielowątkowych mikroserwisów bez konieczności używania dodatkowych frameworków do budowy aplikacji webowych. Dodatkowo, ekosystem Go zapewnia skuteczną integrację z interfejsem Kubernetesa, co jest kluczowe przy zbieraniu logów z klastra komputerowego zarządzanego przez Kubernetes.

## 3.4 Java

Dojrzały i popularny język programowania, który dzięki bogatej dokumentacji i licznej społeczności znacząco przyspiesza proces rozwoju oprogramowania.

## 3.5 Spring Boot

Popularny framework backendowy przeznaczony do budowy aplikacji webowych w architekturze REST. Oferuje sprawdzone rozwiązania w zakresie bezpieczeństwa, routingu oraz mapowania obiektowo-relacyjnego (ORM).

## 3.6 Typescript

Język programowania, rozwijający język JavaScript o dodatkową składnie. Wprowadzone modyfikacje pozwalają na uniknięcie błędów związanych z brakiem silnego typowania.

## 3.7 React

Framework frontendowy użyty do stworzenia klienta aplikacji. React jest najszerzej wspieranym frameworkiem do tworzenia aplikacji w architekturze SPA. To pozwala na znacznie łatwiejsze zarządzanie stanem aplikacji.

## 3.8 Sass

Rozszerzenie klasycznego CSS, które ubogaca podstawową składnie o funkcjonalności minimalizujące duplikacje kodu, poprawiając przy tym czytelność pliku.

## 3.9 Vite

Nowoczesny narzędzie do budowania frontendowych aplikacji webowych, które oferuje szybkie ładowanie modułów podczas rozwoju oraz efektywne budowanie w środowiskach produkcyjnych.

## 3.10 PostgreSQL

Relacyjna baza danych, która została użyta do przechowywania informacji związanych z ustawieniami użytkownika oraz informacji o monitorowanym systemie, które cechują się możliwości ich normalizacji.

### 3.11 MongoDB

Dokumentowa baza danych, w której zostaną przechowywane wygenerowane raporty w języku natural-nym. Raporty takie są długimi dokumentami, które nie wymagają spójności w każdym momencie oraz które ciężko byłoby efektywnie przechowywać i przetwarzać w niedokumentowej bazie danych.

### 3.12 Kafka

jedna z najpopularniejszych platform do strumieniowego przetwarzania danych i kolejkowania zdarzeń. Jej zastosowanie pozwala na uniezależnienie działania mikroserwisów od siebie, zapewniając efektywną komunikację pomiędzy nimi.

### 3.13 ElasticSearch

Nierelacyjna, łatwo skalowalna baza danych, która stała się biznesowym standardem do przechowywania logów.

### 3.14 Redis

Szybka, nierelacyjna baza danych typu klucz-wartość, używana w projekcie jako mechanizm pamięci po-dręcznej, co przyspiesza dostęp do często wykorzystywanych danych oraz zmniejsza obciążenie głównych baz danych.

### 3.15 Nginx

Reverse proxy i serwer webowy, który wspiera aplikację w obsłudze ruchu sieciowego, zwiększając jej skalowalność i wydajność.

## 4 WYNIKI

Jednym z kluczowych ryzyk projektu były koszty oraz jakość generowanych raportów przy użyciu zewnętrznego modelu językowego. Analizy wykazały, że koszt wygenerowania raportu pozostaje na akceptowalnym poziomie – dla pliku logów o wielkości 13 MB wynosi średnio 0,11 USD. Jednocześnie, jakość generowanych raportów uzasadnia realizację projektu. Rozwiązanie skutecznie identyfikuje incydenty i podaje źródła na podstawie których zostały wykryte.

Przykładem może być wygenerowany raport, który zawierał niekrytyczny incydent o brakującym pliku konfiguracyjnym usługi DNS klastra Kubernetes, co oznacza niepoprawną konfigurację. W konsekwencji jest to informacja, która w klasycznym podejściu do monitorowania logów mógłaby zostać pominięta, a sugeruje błąd w konfiguracji, co w przyszłości może doprowadzić do nieprzewidzianego zachowania aplikacji.

Prezentowane podejście umożliwia również regularne monitorowanie klastrów komputerowych oraz wczesne wykrywanie incydentów z ograniczoną ingerencją człowieka.

System zaprojektowano w architekturze mikroserwisowej, aby umożliwić analizę dużego wolumenu danych z wielu różnych źródeł. Architektura ta cechuje się wysoką skalowalnością, elastycznością i odpornością na awarie. Cechy te umożliwiają łatwe skalowanie rozwiązania wraz ze wzrostem liczby danych, ich źródeł oraz klientów.

Testy wydajnościowe wykazały, że system dobrze radzi sobie z plikami logów o rozmiarze większym niż 200 MB przy niewielkim wykorzystaniu zasobów, za sprawą wykorzystania języka Golang. Wydajne wykorzystanie zasobów sprawia, że system można uruchomić również w warunkach ograniczonej dostępności zasobów obliczeniowych.

W celu optymalizacji zarówno kosztów, jak i jakości raportów wprowadzono trzy kluczowe parametry konfiguracyjne:

- **Dokładność** - określa dokładność raportu. Im wyższa dokładność raportu, tym więcej logów zostanie bezpośrednio wykorzystanych do wygenerowania raportów.

- **Okres** – określa przedział czasu, z którego logi są analizowane. Dłuższe okresy wiążą się z wyższymi kosztami przetwarzania.

- **Źródła logów** – obejmuje liczbę węzłów Kubernetesa i aplikacji uwzględnianych w generacji ra-portu. Większa liczba źródeł danych zwiększa koszty analizy.

Dodatkowo, aby zwiększyć wygodę użytkowników, zaimplementowano następujące funkcjonalności:

- **Automatyczne raporty cykliczne** – system umożliwia generowanie raportów w określonych przedziałach czasowych z automatycznym wysyłaniem powiadomień na wybrane kanały komunikacyjne (e-mail, Discord, Slack).

- **Monitorowanie stanu klastra w czasie rzeczywistym** - podczas konfigurowania system prezentuje aplikacje i hosty obecnie działające w klastrze oraz te, które zostały z niego usunięte.

- **Rekomendacje działań** – raporty zawierają nie tylko szczegółowe informacje o incydentach, ale także sugerowane kroki naprawcze, co wspiera administratorów w szybkim reagowaniu na problemy. Administratorzy mają również dostęp do logów źródłowych, co pozwala na głębszą analizę i precyzyjną diagnozę problemów.

- **Dostosowywanie interpretacji logów** - system umożliwiają konfigurację indywidualnego wejścia do modelu językowego dla różnych aplikacji i hostów. Pozwala to na precyzowanie przez administratora dodatkowych uwag odnośnie interpretacji logów i rekomendowania rozwiązań.

Projekt Magpie Monitor udowodnił, że może być elastycznym, wydajnym i ekonomicznym narzędziem, które realnie wspiera administratorów w zarządzaniu złożonymi systemami IT.

# 5 WNIOSKI

Zaimplementowany system potwierdza, że wykorzystanie dużego modelu językowego do analizy logów w klastrach komputerowych, pomimo związanych z tym kosztów, jest efektywne w monitorowaniu i utrzymaniu infrastruktury IT. Średni koszt generowania raportu okazał się wystarczająco niski, aby uzasadnić zastosowanie tego rozwiązania, a jednocześnie zapewnił znaczące korzyści w automatyzacji monitorowania i analizy systemów.

Najważniejszym osiągnięciem projektu jest opracowanie skalowalnego i elastycznego systemu monitorującego, który:

- Automatycznie identyfikuje kluczowe incydenty, ich źródła (konkretne logi), czas ich trwania oraz dostarcza raporty w czytelnej formie.

- Pozwala administratorom skupić się na najistotniejszych problemach dzięki inteligentnej klasyfikacji logów i rekomendacjom działań naprawczych.

- Zapewnia intuicyjny interfejs użytkownika oraz integrację z popularnymi kanałami komunikacyjnymi, zwiększając wygodę korzystania z narzędzia.

Głównym sukcesem projektu jest opracowanie narzędzia, które łączy wydajność technologiczną z praktycznymi korzyściami biznesowymi. System umożliwia:

- Redukcję czasu przestojów, dzięki szybkiemu wykrywaniu i diagnozowaniu problemów. Minimalizację kosztów administracyjnych poprzez automatyzację analizy logów i uproszczenie procesu monitorowania.

- Pozwala administratorom skupić się na najistotniejszych problemach dzięki inteligentnej klasyfikacji logów i rekomendacjom działań naprawczych.

- Zwiększenie niezawodności infrastruktury IT, dzięki możliwości wczesnego reagowania na potencjalne problemy.

Projekt wykazuje znaczący potencjał wdrożeniowy, oferując rozwiązanie, które nie tylko spełnia współczesne wymagania techniczne, ale również przyczynia się do obniżenia kosztów i zwiększenia efektywności zarządzania infrastrukturą IT.

# 6 KIERUNKI ROZWOJU

Kolejnym etapem rozwoju systemu jest implementacja funkcjonalności pozwalającej użytkownikom na wybór modelu językowego wykorzystywanego do generowania raportów, zamiast ograniczenia się wyłącznie do modelu GPT-4.o-mini firmy OpenAI. Szczególnie wartościowym rozwiązaniem będzie przetrenowanie własnego modelu opartego na otwartoźródłowych technologiach, takich jak model Llama od firmy Meta. Takie podejście pozwoli znacząco obniżyć koszty korzystania z systemu, a jednocześnie uniezależnić go od zewnętrznych dostawców, których polityka cenowa może ulec zmianie. Dodatkowo, przetrenowany

na podstawie logów systemowych własny model zapewni wyższą jakość raportów i lepszą adaptację do specyficznych potrzeb użytkowników.

Innym potencjalnym usprawnieniem jest rozwinięcie mechanizmu odfiltrowywania logów poprzez zastosowanie algorytmów analizujących ich znaczenie semantyczne. Dzięki temu, system będzie bardziej precyzyjnie eliminować logi, które nie mają istotnej wartości dla monitorowania systemu. Tego rodzaju ulepszenie wpłynie pozytywnie zarówno na jakość generowanych raportów, jak i na obniżenie kosztów przetwarzania danych, ponieważ zmniejszy się liczba logów analizowanych przez model językowy. W efekcie rozwiązanie stałoby się bardziej wydajne i ekonomiczne, odpowiadając na rosnące potrzeby użytkowników w zakresie monitorowania dużych systemów IT.

Planowany kierunek rozwoju zakłada zastąpienie jednego dużego modelu językowego kilkoma mniejszymi modelami statystycznymi, z których każdy będzie odpowiedzialny za konkretną część procesu analizy logów. Dzięki temu poszczególne modele zostaną wyspecjalizowane w takich zadaniach jak filtrowanie logów, wykrywanie incydentów czy rekomendowanie odpowiednich działań. Taka zmiana pozwoli poprawić jakość wygenerowanych raportów.

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
