# OpenReview forum: "System raportowania zdarzeń systemowych w języku naturalnym z wykorzystaniem dużych modeli językowych"
_pwr.edu.pl/Wrocław_University_of_Science_and_Technology/2024/ZPI_Day — Wrocław University of Science and Technology 2024 ZPI Day Submission_

### Official Review · Reviewer_3AG8 · 2024-12-03
**Abstrakt został starannie przygotowany pod względem strukturalnym i językowym, jednak brakuje szczegółowych opisów niektórych części projektu**

**Confidence:** 3
**Significance Of Results:** 4
**Overall Quality:** 4

**Compliance With Template:**

4: High Quality – The article contains all the required sections, which are well-written and substantively correct, although minor errors or shortcomings may be present. The overall structure is clear and coherent.

**Description Of Results:**

4: High Quality – The results are described in detail and supported by usage examples or evaluations. The description is reliable but may lack full depth of analysis.

**Feedback On Consistency:**

Praca napisana spójnie, choć w wielu miejscach warto było dodać szczegóły techniczne, np. dodanie konkretnych funkcjonalności, wyjaśnienie mechanizmu rekomendacji działań dla administratorów. Warto byłoby również zaznaczyć wpływ modelu językowego na wynik.

**Potential For Development:**

Wymienione w pracy kierunki dalszych prac wskazują na realistyczne podejście do tematu.

**Project Nature Evaluation:**

Projekt wykazuje cechy pracy inżynierskiej. Opisane aspekty pracy wskazują na realizację założonego celu.

**Technical Language Precision:**

4: High Quality – The language is appropriate for a technical report. Terminology is used correctly, and statements are precise, with only minor shortcomings that do not affect the overall clarity.

---

### Official Review · Reviewer_3wQm · 2024-12-04
**Recenzja systemu Magpie Monitor**

**Confidence:** 5
**Significance Of Results:** 5
**Overall Quality:** 5

**Compliance With Template:**

5: Very High Quality – The article contains all the required sections, which are written in a very detailed, clear, and error-free manner. The structure is professional and meets expectations, and the content adheres to the highest substantive and formal standards.

**Description Of Results:**

5: Very High Quality – The results are described in detail, clearly and comprehensively, supported by thorough evaluation, analysis, and convincing usage examples. The description meets the highest substantive standards.

**Feedback On Consistency:**

Artykuł przedstawia projekt w sposób spójny i logiczny, jasno definiując problem oraz proponowane rozwiązanie. Problem manualnej analizy logów, czasochłonnej i podatnej na błędy, został precyzyjnie zidentyfikowany. Artykuł opisuje system Magpie Monitor jako narzędzia do automatyzacji tego procesu, z zastosowaniem dużych modeli językowych do generowania raportów w języku naturalnym. Wnioski podsumowują osiągnięcia, podkreślając skuteczność systemu w minimalizacji ryzyka przestojów oraz redukcji obciążenia administratorów IT.

**Potential For Development:**

System może zostać wzbogacony o możliwość wyboru i dostosowywania modeli językowych przez użytkowników. Planowane zastąpienie dużego modelu językowego (w tym momencie jest to model GPT-4o) przez mniejsze, wyspecjalizowane modele (nauczone na danych dostarczanych bezpośrednio przez klientów) stanowi ciekawą propozycję, mogącą zwiększyć dokładność uzyskiwanych wyników.

**Project Nature Evaluation:**

Artykuł dobrze przedstawia założenia projektu i osiągnięte rezultaty. Projekt Magpie Monitor wykazuje znaczny potencjał praktyczny, oferując skalowalne, elastyczne i innowacyjne narzędzie wpasowujące się w zestaw istniejących narzędzi do zarządzania infrastrukturą IT:
- System adresuje kluczowe potrzeby w zarządzaniu infrastrukturą IT, w tym automatyczne raportowanie i wczesne wykrywanie problemów.
- Wykorzystanie technologii takich jak Kubernetes, Docker, Golang, Elasticsearch, czy MongoDB, w połączeniu z dużymi modelami językowymi, wskazuje na wysoki poziom zaawansowania technicznego projektu.
- Architektura mikroserwisowa zapewnia skalowalność i elastyczność, a intuicyjny interfejs użytkownika zapewnia łatwość użytkowania systemu.

**Technical Language Precision:**

5: Very High Quality – The language is entirely appropriate for a technical report. All terms are used correctly and precisely, and the style is professional, clear, and coherent, without any errors or ambiguities.

---

### Official Review · Reviewer_1ys1 · 2024-12-06
**Krótki opis systemu analizy logów zdarzeń systemowych z wykorzystaniem LLM'ów**

**Confidence:** 3
**Significance Of Results:** 5
**Overall Quality:** 5

**Compliance With Template:**

4: High Quality – The article contains all the required sections, which are well-written and substantively correct, although minor errors or shortcomings may be present. The overall structure is clear and coherent.

**Description Of Results:**

5: Very High Quality – The results are described in detail, clearly and comprehensively, supported by thorough evaluation, analysis, and convincing usage examples. The description meets the highest substantive standards.

**Feedback On Consistency:**

System rozwiązuje konkretny problem – automatyzuje analizę dużych zbiorów logów, poprawiając efektywność pracy administratorów.

**Potential For Development:**

Kierunki dalszego rozwoju, takie jak:

Wdrożenie algorytmów semantycznej analizy logów, co zwiększy precyzję raportowania.
Zastosowanie specjalizowanych modeli statystycznych do różnych etapów analizy, co poprawi efektywność i jakość generowanych raportów.

**Project Nature Evaluation:**

Zastosowania metod technicznych: Wykorzystano szereg technologii, takich jak Kubernetes, Docker, Golang, oraz mikroserwisową architekturę, co zapewnia wysoką skalowalność i wydajność.
Rozwiązań technologicznych: Implementacja zaawansowanych modeli językowych w celu generowania raportów w języku naturalnym stanowi innowacyjne podejście, które odróżnia ten projekt od istniejących rozwiązań.

**Technical Language Precision:**

4: High Quality – The language is appropriate for a technical report. Terminology is used correctly, and statements are precise, with only minor shortcomings that do not affect the overall clarity.

---

### Decision · Program_Chairs · 2024-12-10

Accept (Poster)